



# Measuring turbulent $CO_2$ fluxes with a closed-path gas analyzer in marine environment

Martti Honkanen[1], Juha-Pekka Tuovinen[2], Tuomas Laurila[2], Timo Mäkelä[2], Juha Hatakka[2], Sami Kielosto[1,3], and Lauri Laakso[1,4]

[1]Meteorological and Marine Research Programme, Finnish Meteorological Institute, Finland
[2]Climate Research Programme, Finnish Meteorological Institute, Finland
[3]Marine Ecology Research Laboratory, Finnish Environment Institute, Finland
[4]School of Physical and Chemical Sciences, North-West University, Potchefstroom Campus, South Africa

**Correspondence:** Martti Honkanen (martti.honkanen@fmi.fi)

**Abstract.** Sea-air fluxes of carbon dioxide ($CO_2$) were measured using the eddy covariance method at a new station established on the Utö island in the Baltic Sea. The flux measurement system is based on a closed-path infrared gas analyzer (LI-7000, LI-COR) requiring only occasional maintenance, so the station is capable of continuous monitoring. However, such infrared gas analyzers are prone to significant water vapor interference in a marine environment, where $CO_2$ fluxes are small.

In July–October 2017, two LI-7000 analyzers were run in parallel to test the effect of a sample air drier which dampens water vapor fluctuations, and a virtual impactor, included to remove liquid sea spray, both of which were attached to the sample air tubing of one of the analyzers. The systems showed closely similar ($R^2 = 0.99$) sea-air $CO_2$ fluxes when the latent heat flux was low, which proved that neither the drier nor the virtual impactor perturbed the $CO_2$ flux measurement. However, the undried measurement had a positive bias that increased with increasing latent heat flux, suggesting water vapor interference.

For both systems, cospectral densities between vertical wind speed and $CO_2$ were distributed within the expected frequency range, with a moderate attenuation of high-frequency fluctuations. While the setup equipped with a drier and a virtual impactor generated a slightly higher flux loss, we opt for this alternative for its reduced water vapor cross-sensitivity and better protection against sea spray. The integral turbulence characteristics were found to agree with the universal stability dependence observed over land. Non-stationary flow conditions caused unphysical results, which resulted in a high percentage (up to 63 %)

of discarded measurements. After removing the non-stationary cases, the direction of the sea-air $CO_2$ fluxes was in good accordance with the measured $CO_2$ partial pressure difference between the sea and the atmosphere. Atmospheric $CO_2$ concentration changes larger than 2 ppm during a 30 min averaging period were found to be associated with the non-stationarity of $CO_2$ fluxes.

The Utö Atmospheric and Marine Research Station continues to monitor the regional $CO_2$ exchange between the sea and

the atmosphere, utilizing the results of this work.



## 1 Introduction

Anthropogenic actions, such as combustion of fossil fuels and land use changes, have perturbed the global carbon cycle, resulting in climatic changes (e.g. Solomon et al., 2009). A quarter of the anthropogenic carbon dioxide ($CO_2$) emissions to the atmosphere are bound by the oceans (Heinze et al., 2015), causing ocean acidification (Feely et al., 2009). To better

understand the global carbon cycle, measurements of the $CO_2$ exchange between the atmosphere and marine ecosystems are required. These measurements are also useful for developing parameterizations of gas exchange intensity, such as the gas transfer velocity, used in global carbon models (e.g. Takahashi et al., 2002). In the case of coastal seas, sea-air $CO_2$ flux measurements provide information about the feedbacks between the aquatic carbonate system and marine ecosystems, since the direction and magnitude of these fluxes depend on the photosynthetic carbon assimilation in surface seawater. Although

the continental margin seas cover only a small portion of the oceans, up to 15 % of the total ocean primary production takes place in these seas, which are responsible for over 40 % of the total oceanic carbon sequestration (Muller-Karger et al., 2005).

With the development of fast-response infrared $CO_2$ analyzers, it has become possible to apply the eddy covariance method for directly measuring the sea-air $CO_2$ fluxes (Jones and Smith, 1977). Early on, however, Webb et al. (1980) showed that the $CO_2$ flux measurements made with this technique need to be corrected for the effects of temperature and water vapor ($H_2O$)

fluctuations. This correction can be significant in aquatic environment, even of the same order as the measured $CO_2$ flux (Sahlée et al., 2011). More recently, it has been recognized that $CO_2$ gas analyzers suffer from water vapor cross-sensitivity (Kohsiek, 2000). Blomquist et al. (2014) concluded that this is the most significant error source in sea-air $CO_2$ flux measurements, because the $CO_2$ fluxes in marine environment are typically small, as compared to terrestrial ecosystems. A solution to the cross-sensitivity problem is to dry the sample air before the measurement (e.g. Miller et al., 2010). Also, efforts have been

made to correct for the cross-sensitivity problem in the data post-processing phase (e.g. Prytherch et al., 2010; Edson et al., 2011). However, these corrections may be inadequate and thus do not obviate the use of a drier (Landwehr et al., 2014).

The cross-sensitivity results from the overlap of the infrared radiation frequency bands of $CO_2$ and $H_2O$, because of which the $H_2O$ molecules present will increase the apparent $CO_2$ concentration. If the optical filter, used for the selection of the transmitted frequency band, leaks out-of-band radiation, a substance with a different absorption frequency band can interfere

the measurement. This effect is referred to as the direct absorption cross-sensitivity. In addition, the collisions with different molecules cause the frequency bands to broaden (so-called pressure broadening). By testing some commercial infrared gas analyzers, Kondo et al. (2014) found out that the factory-calibrated correction for the direct absorption interference may not be optimized and that the pressure broadening effect caused an overestimation of the $CO_2$ flux, which increased with increasing water vapor flux.

Infrared gas analyzers are classified according to the type of the optical path: the open-path analyzers measure the absorption of the infrared signal in ambient air, whereas the closed-path gas analyzers have an enclosed measurement chamber. Both types suffer from $H_2O$ cross-sensitivity but otherwise have differing pros and cons; for instance, the closed-path instruments are known to act as low-pass filters, which generates a loss in the measured flux (Leuning and King, 1992). On the other hand, a long sample line attenuates temperature fluctuations, thus eliminating the need for correcting for sample air expan-



sion/contraction (Rannik et al., 1997). For a closed-path analyzer, the dilution due to water vapor (Webb et al., 1980) can be corrected accurately as a point-by-point operation on the high frequency data, which is not possible with open-path analyzers (Ibrom et al., 2007).

As direct sea-air gas exchange measurements are performed in the surface boundary layer, in a close proximity to the water surface, liquid sea spray may block the optical path of open-path sensors and clog the inlet of closed-path analyzers. In addition, high relative humidity can produce $H_2O$ condensation on the lenses of an open-path system. In a closed-path system, the accumulation of sea salt on the optical lenses is minimized as the inlet is typically protected by a Teflon filter. Efforts have been made to solve the sea-spray contamination problem of open-path sensors by cleaning the optics regularly (Kondo and Tsukamoto, 2007). However, this may be technically challenging, because sea salt films can be formed on the windows of an open-path analyzer in a matter of hours (Miller et al., 2010).

Open-path gas analyzers have been mostly applied for sea-air $CO_2$ flux measurements due to their low power consumption, small high frequency attenuation and ease of data synchronization with wind measurements (Blomquist et al., 2014). Additionally, Miller et al. (2010) found out that closed-path sensors are more sensitive to (ship) motion than the open-path sensors.

While the eddy covariance method is widely used for directly measuring the surface-atmosphere exchange of energy and matter, it is based on several theoretical assumptions. These include the horizontal homogeneity of terrain and the stationarity of transport processes, and that turbulence is fully developed and there exists no other transport mechanisms than vertical turbulence (e.g. Dabberdt et al., 1993). Foken and Wichura (1996) noted that flow non-stationarity is one of the most serious problems affecting the surface exchange measurements, as in such conditions the observed turbulent flux does not equal the flux at the surface. The stationarity assumption can be violated owing to diurnal forcings and varying weather patterns, for example.

The Baltic Sea forms a large and diverse biogeochemical system, providing a great potential for studying interactions between the marine ecosystem and the aquatic carbonate system. However, only a few fixed measurement sites measuring sea-air $CO_2$ fluxes are located in the Baltic Sea (Rutgersson et al., 2008; Lammert-Stockschaeder and Ament, 2015). A micrometeorological tower placed on the shore of an island offers a cost-effective alternative to ship measurements, as maintenance and installations are more effortless. Moreover, a motion correction, required with ship measurements, is not needed and the flow distortion can be minimized with a suitable positioning of the flux tower.

In this paper, we introduce a newly established and currently operating eddy covariance measurement site, located on the Utö island in the Baltic Sea, and analyze empirically the effect of water vapor on the $CO_2$ sea-air fluxes. After experimenting with an open-path and a semi-open-path gas analyzer, we opted for a closed-path sensor, which can be protected against sea spray contamination, and the drying of sample air is straightforward to implement. We compare measurements made with two identical closed-path analyzers, one of which is equipped with a drier and a virtual impactor. We also address the quality of these measurements by analyzing the stationarity and integral turbulence characteristics of the flow and the homogeneity of flux footprints.





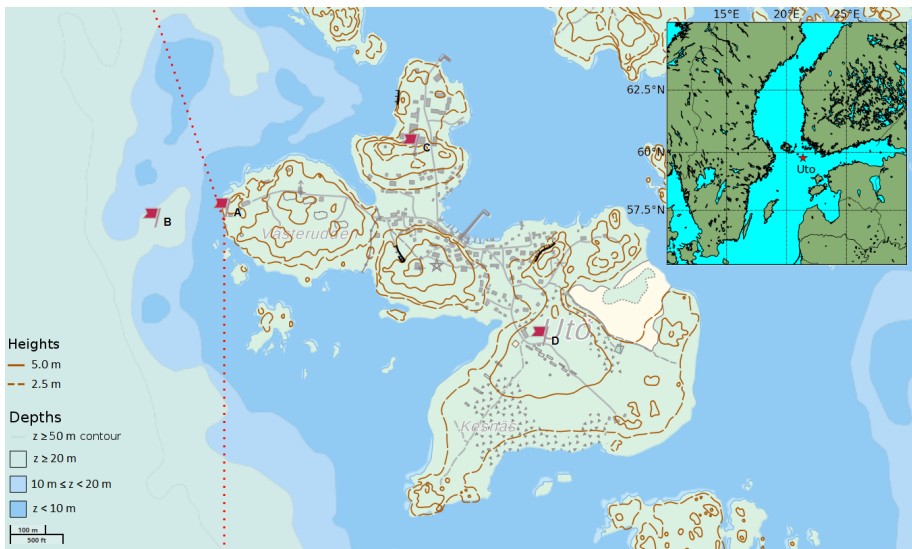

**Figure 1.** Location of the Utö island in the Archipelago Sea. The research installations on the island consist of the Utö Atmospheric and Marine Research Station (A), a flow-through pumping system (inlet at B), the Atmospheric ICOS station (C), and a weather and air quality station (D). The red dashed lines indicate the wind sector suitable for sea-air flux measurements (180–340°).

## 2 Materials and methods

### 2.1 Site description

The measurement site is located on the island of Utö in the southern edge of the Archipelago Sea in the Baltic Sea (59°46'55"
N, 21°21'27" E) (Fig. 1). The Archipelago Sea is a small sea area between the southwest coast of Finland and the sea of Åland
and is characterized by thousands of small islands. The island of Utö is a treeless cliff with small shrubs, with an area of 0.81
km$^2$ .

Since 2012, the Finnish Meteorological Institute has been building up the new Utö Atmospheric and Marine Research Station
in collaboration with the Finnish Environment Institute. The choice of this island was due to its easy access, well-developed
technical infrastructure, permanent inhabitation and the long-term meteorological and marine measurements. Finnish Meteo-
rological Institute's meteorological measurements date back to 1881, and the seawater temperature and salinity measurements
were initiated in 1900 (Laakso et al., 2018). Also, the Integrated Carbon Observation System (ICOS) research infrastructure
has an atmospheric station on the island.

The Utö Atmospheric and Marine Research Station and its flux tower are located on the western side of the island. To the
west of the shore, the water depth quickly deepens to 80 m. The closest shoal, Tratten, is located 1.4 km to the southwest and
has an area of 0.6 hectares.





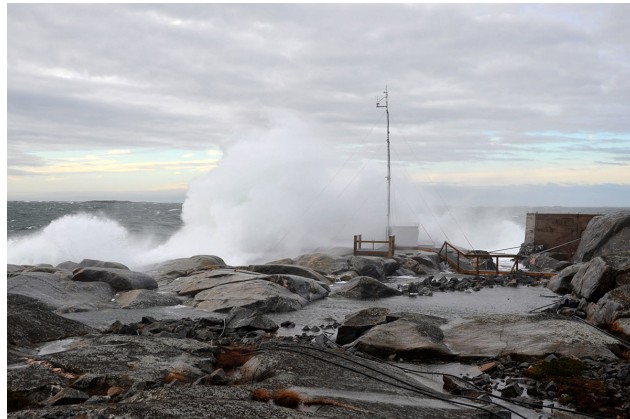

**Figure 2.** The flux tower on the shore of the Utö island during a high western swell (photo by Ismo Willström).

The mean annual wind speed at Utö is $7.1\,\mathrm{m\,s^{-1}}$. On average, the minimum monthly wind speed occurs in July ($5.6\,\mathrm{m\,s^{-1}}$) and the maximum in December ($8.9\,\mathrm{m\,s^{-1}}$). The wind blows approximately $60\,\%$ of the time from the sector between south and northwest (Fig. 1), providing a good temporal coverage for the sea-air flux measurements. (Pirinen et al., 2012)

The mean annual air temperature in Utö is $6.6\,°\mathrm{C}$, whereas the mean annual surface seawater temperature at the depth of

$5\,\mathrm{m}$ is $8.1\,°\mathrm{C}$. In deep water (at $50\,\mathrm{m}$ depth), the mean annual seawater temperature is $4.4\,°\mathrm{C}$. The maximum monthly mean air temperature ($16.7\,°\mathrm{C}$) is observed in July, whereas the seawater at $5\,\mathrm{m}$ depth reaches its maximum temperature ($16.1\,°\mathrm{C}$) in August. The minimum temperature of air ($-2.1\,°\mathrm{C}$) occurs in February, while the temperature of the surface seawater is at its minimum ($1.0\,°\mathrm{C}$) in March. On average, there exists ice around Utö for every few years with a typical ice cover duration ranging from one to three months. (Laakso et al., 2018)

Unlike the oceans, the carbon cycle in the Baltic sea is heavily influenced by biological activity. During summer, the partial pressure of $CO_2$ ($pCO_2$) in surface water declines to approximately $15\,\mathrm{Pa}$ as a result of primary production. $pCO_2$ in surface water has its maximum ($60\,\mathrm{Pa}$) during winter, when biological activity is diminished, mineralization prevails and mixing processes transport $CO_2$-rich seawater to the surface. Due to the annual cycle of primary production, the Baltic Sea is a source of $CO_2$ to the atmosphere in winter and a sink in summer. Upwelling and diurnal biological cycle generate short-term variations

in the surface water $pCO_2$. (Wesslander, 2011)

## 2.2  Instrumentation

A 9 m tall micrometerological tower is placed on the western shore of the island (Figs. 1, 2). The tower is mounted on a cliff that has no vegetation. The base of the tower is approximately $3\,\mathrm{m}$ above the sea level and the horizontal distance between the tower and the sea is approximately $4\,\mathrm{m}$. When measuring sea-air exchange, only the fluxes related to the westerly winds

(180–340°) are used.



Air velocity components together with air temperature, $T$, were measured with an acoustic anemometer/thermometer (uSonic-3, Metek) attached to the top of the tower (11.5 m a.s.l.). $CO_2$ and $H_2O$ molar fractions were measured by using two closed-path differential non-dispersive infrared gas analyzers (LI-7000, LI-COR) that were placed in a 1.5 m tall instrument hut close to the tower (Fig. 2). The inlets of both sample flows are protected with a grate and are located directly beneath the anemometer; the

distance between the inlets and the lower anemometer transducers is 30 cm. The outside parts of the tubes are approximately 10 m long and are made of Teflon (inner diameter of 3.175 mm) and steel (inner diameter of 4.0 mm). The steel-tube sample line is equipped with a virtual impactor (see Appendix), to protect the instrument from possible exposure to liquid water, and a 30 cm long PD-100T-12-MKA drier (Perma-Pure) to attenuate water vapor fluctuations. The drier is based on the partial pressure difference that drives water vapor from the sample air to the purge stream through a Nafion membrane. Nafion driers

are found to have only a small permeability to $CO_2$ (Welp et al., 2013).

Inside the instrument hut, the flow continues in 1 m long Bev-A-Line tubes, which are protected by Teflon filters and connected to the analyzers. Data are logged at a 10 Hz frequency and transmitted through the Internet to a server. The hut temperature is regulated using a fan and a radiator, since large temperature changes influence the performance of the $CO_2$ analyzers.

An external pump is used for producing a sample flow at a rate of approximately $6 \, l \, min^{-1}$ for both setups. We operate the LI-7000 gas analyzers in the absolute mode, which means that a small stream of zero gas (0 ppm $CO_2$) is constantly flowing through the reference cell of both analyzers. The gas analyzers are calibrated with zero and span (364.4 ppm $CO_2$) gases every three months. After calibration, the Teflon filters are renewed and the virtual impactor is cleaned. The outside tubes are washed with an isopropyl alcohol-water mixture annually. At the same time, the Bev-A-Line tubes and the internal chemicals of the

gas analyzers are renewed and the optical paths are cleaned.

The setup with the Nafion drier and the virtual impactor is referred here to as the 'test setup' and the other setup as the 'standard setup'. The standard setup represents a typical closed-path gas analyzer-based eddy covariance setup commonly applied on terrestrial ecosystems (e.g. Aurela et al., 2015). The test setup is an improved version of this configuration, designed especially for a marine environment where water vapor and sea spray are likely to give rise to problems.

## 25 2.3 Methods used in flux calculations

As the LI-7000 gas analyzer does not directly provide molar fractions with respect to dry air, we first calculated the dilution corrected $CO_2$ molar fraction as:

$$c = \frac{c_w}{1 - q},\qquad(1)$$

where $c_w$ is the uncorrected molar fraction of $CO_2$ and $q$ is the molar fraction of $H_2O$.

It is assumed that the long pipe lines, both Teflon and steel, strongly attenuate the temperature fluctuations and thus there is no need for correcting for expansion/contraction effects in the sample air (Rannik et al., 1997). The eddy covariance mass flux was calculated from the fluctuations of molar fractions of an atmospheric quantity, $\chi$, and vertical wind speed, $w$:

$$F = \rho_d \overline{w'\chi'},\qquad(2)$$





where $\rho_d$ is the dry air density. The primes denote the $10\,\text{Hz}$ fluctuations with respect to a time average and the overbar denotes arithmetic averaging. An averaging period of $30\,\text{min}$ was adopted. For each period, the time delay between the acoustic anemometer and gas analyzer data flows was determined by maximizing the (absolute) covariance in Eq. 2 within a predefined time window, and a double coordinate rotation was applied to ensure zero vertical wind speed.

Obvious outliers were discarded by including only those $CO_2$ fluxes that are within three standard deviations from the mean. The fulfillment of several theoretical requirements for the eddy covariance measurements was considered. The non-stationarity of the flux measurements was analyzed by comparing the average of the $CO_2$ fluxes calculated for $5\,\text{min}$ subperiods ($\frac{1}{6}\sum_{i=1}^{6} \overline{w'c'}_{5\,\text{min, i}}$, where $i$ is the index of the subperiod) with the $30\,\text{min}$ flux ($\overline{w'c'}_{30\,\text{min}}$) (Foken and Wichura, 1996). The relative non-stationarity, $RN$, for a given variable, $\zeta$, was defined as:

$$RN_\zeta = \left| \frac{\frac{1}{6}\sum_{i=1}^{6} \zeta_{5\,\text{min, i}} - \zeta_{30\,\text{min}}}{\zeta_{30\,\text{min}}} \right|. \tag{3}$$

In the case of $CO_2$ flux, $\zeta = \overline{w'c'}$. To test the development of turbulence, integral turbulent characteristics of temperature and vertical velocity were compared to universal surface-layer functions (Thomas and Foken, 2002). It is assumed that turbulent transport of all scalars ($T$, $CO_2$ and $H_2O$) are similar, and thus only the characteristics of $T$ are examined here. For this examination, only stationary flow conditions ($RN_{\overline{w'c'}} < 0.3$) within the sea sector are included. An observation is also discarded if

a positive momentum flux is measured, as these can be considered a sign of flow non-stationarity (Yang et al., 2016).

### 2.3.1   Spectral analysis

High frequency attenuation was examined by calculating the cospectra between $w$ and $c$ ($Co_{wCO_2}$) and $q$ ($Co_{wH_2O}$) and comparing these with the corresponding cospectra between $w$ and $T$ ($Co_{wT}$). For this purpose, it is assumed that the high frequency end of $Co_{wT}$ is unattenuated and the atmospheric turbulent transport of heat, water vapor and $CO_2$ are similar.

Before calculating the half-hourly cospectra using the discrete Fourier transfrom, the $10\,\text{Hz}$ data were linearly detrended and the Hamming window was applied. The cospectra were normalized with the corresponding covariances and interpolated into 64 logarithmically equally spaced bins.

Observations with an appropriate wind direction (180–340°) and stationary flow conditions, based on the criterion that $RN_{\overline{w'c'}} < 0.3$, were accepted for spectral analysis. Observations during slightly unstable to slightly stable hydrostatic stability

conditions ($|z/L| < 0.05$) were examined, where $z$ is the measurement height and $L$ is the Obukhov length. In total, 612 observations met these criteria. Furthermore, every cospectrum was visually inspected and the clearly distorted ones (365) were discarded. Thus, 247 observations from a 4 month period were used for the spectral analysis.

To describe the spectral attenuation of the flux as a function of frequency, $f$, a transfer function, $\Gamma$, was determined. For fitting the transfer function, the cospectra were normalized so that their peaks were leveled. Also, the outermost points were

discarded. Using a non-linear least squares fit, an exponential transfer function was fitted to the ratio of $Co_{wCO_2}$ or $Co_{wH_2O}$ to $Co_{wT}$:

$$\Gamma(f) = \exp\left[-\log(2)\left(\frac{f}{f_0}\right)^2\right], \tag{4}$$



where $f_0$ is the half-power frequency at which the ratio of cospectral densities is 0.5. In the case of H$_2$O fluxes, a power-law relationship between $f_0$ and relative humidity ($RH$) was additionally derived similarly to Mammarella et al. (2009).

To correct for the high-frequency attenuation of fluxes, which depends on stability and wind speed, the universal $Co_{wT}$ equations reformulated by Horst (1997) from those originally presented by Kaimal et al. (1972) and Kaimal and Finnigan (1994) were used as a reference.

## 2.4 Auxiliary data

In addition to the flux measurements, we present here data of the dissolved CO$_2$ concentration in surface seawater, which is monitored using a SuperCO$_2$ system (Sunburst Sensors) connected to a flow-through pumping system. The SuperCO$_2$ instrument consists of a shower-head equilibration chamber and an infrared gas analyzer (840A, LI-COR). To account for sensor drift, the flows of four reference gases are directed into the instrument every four hours, and a linear correction is calculated from these calibration measurements. The system automatically cleans the equilibration chamber with hydrogen peroxide periodically. The inlet for seawater intake is located 250 m west of the station (Fig. 1) and it is kept at 5 m depth by using floats. The water depth at this location is 23 m. Since sample water exchanges heat with the pipe and its surroundings, a correction for this temperature effect was applied according to Takahashi et al. (1993):

$$pCO_{2_{in}} = pCO_{2_{eq}} \exp[0.0423(T_{in} - T_{eq})], \tag{5}$$

where $pCO_{2_{in}}$ is the temperature-corrected partial pressure of CO$_2$, $pCO_{2_{eq}}$ is the observed partial pressure of CO$_2$, $T_{in}$ is seawater temperature measured close to the inlet, and $T_{eq}$ is the temperature measured in the tubing right before the equilibration chamber. This correction is, however, applied only since the beginning of August 2017, after the installation of the inlet thermometer measuring $T_{in}$. Additionally, sea bottom temperature below the inlet is monitored using another thermometer.

The atmospheric concentration of CO$_2$ at the height of 57 m was measured at the ICOS station (C in Fig. 1). At this station, the dry molar fraction of CO$_2$ is measured by using a very stable cavity ring-down spectroscopy technique (G2401, Picarro). For more information on the atmospheric ICOS measurements at Utö, the reader is referred to Kilkki et al. (2015).

## 3 Results and discussion

### 3.1 Suitability of the site for eddy covariance measurements

#### 3.1.1 Environmental conditions

The measurement period (1 July – 1 November 2017) represents the late summer and autumn seasons of carbon cycle, when biological activity diminishes and the sea-air pCO$_2$ difference shifts from negative to positive (Fig. 3e). At the beginning of this period, from July to mid-August, the atmospheric pCO$_2$ exceeded that in the sea, and the sea acted as a sink of atmospheric CO$_2$. The average sea-air pCO$_2$ difference from July to mid-August was 8.0 Pa. During the latter part of August, the sea turned to a source as a result of diminished primary production. The efflux was enhanced when thermal stratification broke down and





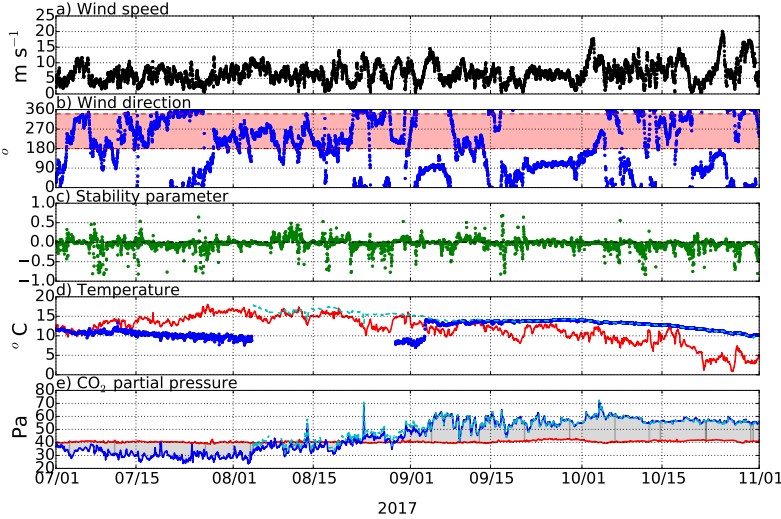

**Figure 3.** Environmental conditions at Utö in July–November 2017: (a) wind speed, (b) wind direction (red band indicates the directions suitable for sea-air flux measurements), (c) stability ($z/L$), (d) air temperature (red) and sea temperature at 5 m depth (cyan) and at the sea bottom (blue), and (e) $CO_2$ partial pressure in air (red), and temperature-corrected (cyan) and uncorrected (blue) $CO_2$ partial pressure in seawater at 5 m depth. Wind speed, wind direction, stability and air temperature were measured in the flux station (A in Fig. 1). Seawater temperatures and $pCO_2$ were measured at the inlet (B in Fig. 1). The $pCO_2$ in air was measured in the ICOS station (C in Fig 1).

$CO_2$-rich water surfaced at the beginning of September, resulting in increased surface $pCO_2$. After this event, the partial pressure difference stayed predominantly within 10–20 Pa for the rest of the measurement period, showing only occasional deviations from this range.

The values reported for the sea surface $pCO_2$ represent conditions at the depth of 5 m at a single point 250 m away from the
5   flux tower, in the middle of the flux measurement sector. In some cases, this location may not represent the carbonate conditions of the whole sea sector area. Especially during the beginning of the measurement period, it is possible that there is horizontal differences in the surface $pCO_2$ due to variations in biological activity (Rutgersson et al., 2008). During other periods, the horizontal differences are likely to be small in the open sea areas and the surface sea layer is likely to be well-mixed at least to the depth of 5 m. Thus, we assume that our $pCO_2$ observations represent the surface conditions.
10   During our study period, the wind blew from the sea sector (180–340°) for 49 % of the time (Fig. 3). This caused gaps in the sea-air flux time series; for instance during the last two weeks of September, the wind directions were unsuitable for sea-air exchange measurements. The average wind speed within the marine wind sector was 6.9 m s$^{-1}$. Stability was mostly near-neutral. A small part (3.7 %) of the momentum fluxes measured in the marine sector were positive, and thus the corresponding $CO_2$ flux data were discarded.



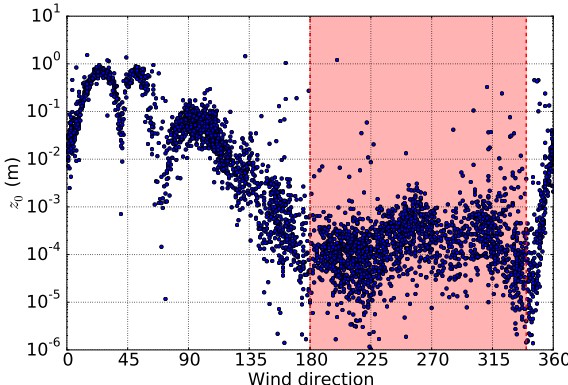

**Figure 4.** Surface roughness length as a function of wind direction. The red band indicates the directions considered the open sea sector.

In July, air temperature was still gradually increasing and it reached 18 °C by the end of the month. From the beginning of August, both air and sea surface temperature decreased approximately linearly. During the last three months, surface sea water cooled by 7 °C and air by 10 °C.

### 3.1.2 Horizontal homogeneity

Horizontal homogeneity within the sea sector (180–340°) was examined in terms of the surface roughness length ($z_0$), which was calculated from the logarithmic wind profile law using data collected during neutral conditions, $|z/L| < 0.01$ (Fig. 4).

The sea sector is clearly visible in a range of 180–340°, where $z_0$ is mainly lower than 1 mm. Outside this sector, $z_0$ is clearly higher, up to 1 m in northeastern wind directions. The sea sector indicated by $z_0$ coincides well with the sector deduced from the geographical map (Fig. 1).

While the sea sector can be considered sufficiently homogeneous, there is scatter in $z_0$ reflecting the fact that it depends on the sea state. Taylor and Yelland (2001) found that over a sea surface $z_0$ depends on the significant wave height and the slope of the wave. Since the measurement site is located on the coast, both the fetch and swell can depend on wind direction. Especially, a large swell can originate from the south, i.e. from the open waters of the Baltic Proper.

### 3.1.3 Spectral characteristics

The cospectra $Co_{wCO_2}$ of both measurement setups agreed well with the modeled cospectrum in all frequencies (Fig. 5). The low frequency ends of the measured cospectra approach zero in a similar way to the modeled cospectrum.

The attenuation of the highest frequencies was slightly higher in the more complex tubing: $Co_{wCO_2}$ of the test setup (with a drier and a virtual impactor) diverged from the modeled $Co_{wT}$ at a slightly lower frequency than that of the standard setup. The half-power frequency for the standard and test system was 0.77 Hz and 0.59 Hz, respectively. These values were used to





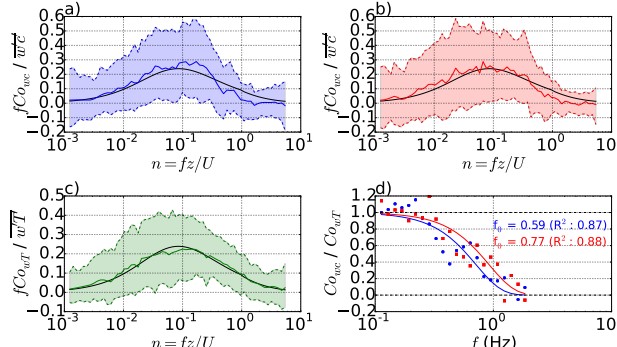

**Figure 5.** Cospectral densities $Co_{wCO_2}$ of (a) the test setup and (b) the standard setup, and (c) $Co_{wT}$. The solid lines represent the mean cospectra and the dashed lines represent the 10th and 90th percentiles. The black line is the model $Co_{wT}$ (Horst, 1997). (d) Ratios of cospectral densities (dots) and the fitted exponential transfer functions (solid lines), where the red color indicates the standard setup and the blue color indicates the test setup.

correct the $CO_2$ fluxes according to the model cospectrum. The high-frequency correction of the $CO_2$ flux during the typical meteorological conditions ($U = 6.9 \, \mathrm{ms}^{-1}$ and $z/L = -0.05$) is 15 % for the test setup and 11 % for the standard setup.

   The drier and virtual impactor added approximately 1 m to the length of tubing, which only has a minor effect on the attenuation of turbulent fluctuations. However, the virtual impactor forms a $90°$ angle to the tubing, which can stabilize the
flow making it laminar at a higher Reynolds number, $Re$, than in a straight tube (Lenschow and Raupach, 1991). It would be ideal to have turbulent conditions in the tubes, because scalar fluctuations dampen less in a turbulent pipe flow due to the more uniform velocity profile (Lenschow and Raupach, 1991). At low temperatures ($0 \, °\mathrm{C}$), $Re$ of the standard setup was calculated to be 2980, whereas in the test setup it was 2380. By using the tube attenuation equations of turbulent flow presented by Massman and Ibrom (2008), the half-power frequencies were 13.7 and 7.2 Hz for the standard and test setup, respectively. Thus, at
low temperatures, fluctuation attenuation due to transport in the tubes is not likely to affect the measurements. However, at higher temperatures ($20 \, °\mathrm{C}$), the Reynolds number of the test setup (2100) falls into the laminar flow region, because kinematic viscosity increases with increasing temperature. By using the tube attenuation equations for laminar flow by Lenschow and Raupach (1991), the half-power frequency of the test setup at $20 \, °\mathrm{C}$ was only 1.4 Hz, which is still higher than the empirically determined $f_0$.
The high frequency attenuation of $H_2O$ signals of the standard setup was corrected in order to accurately compare the $CO_2$ flux measurements with different setups as a function of latent heat flux. The high frequency attenuation of $H_2O$ flux of the standard setup was marked: $f_0$ ranged between 0.04 and 0.22 Hz, decreasing with increasing $RH$, which varied within 48–100 %. This attenuation is caused by sorption and desorption of water vapor at the walls of the tubings (Massman and Ibrom, 2008). During typical humidity conditions (RH = 80 %), the flux attenuation is 44 % of the real $H_2O$ flux. The $Co_{wH_2O}$ of the
test setup was found to be distorted in all frequencies.





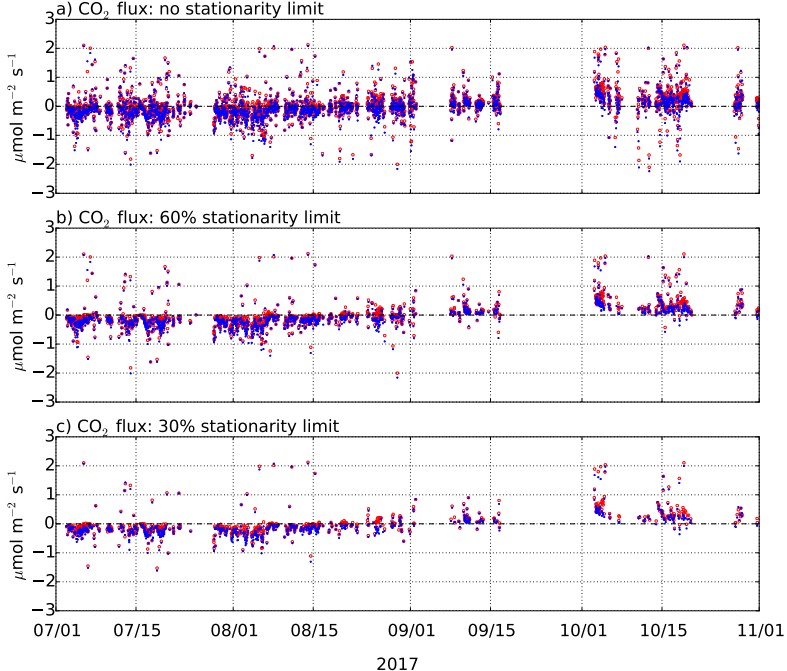

**Figure 6.** The effect of removal of non-stationarity cases on the $CO_2$ flux data: (a) no stationarity limit, (b) $RN_{\overline{w'c'}} < 0.6$ and (c) $RN_{\overline{w'c'}} <$ 0.3. Red circle refers to the standard setup and blue dot to the test setup.

### 3.1.4 Stationarity

About half of the flow conditions for which the $CO_2$ flux was determined, 55 % for the standard setup and 63 % for the test setup, were found to be non-stationary, i.e. the difference between the 30 min flux and the mean of the corresponding 5 min fluxes exceeded 30%.

5     We found that this stationarity criterion removed unphysical values effectively (Fig. 6). During July, the partial pressure difference was negative (-9.3 Pa on average), and thus downward (negative) fluxes were expected. Relaxing the stationary limit from 30 % to 60 % had only a small effect on data filtering. If no stationarity limit was applied, 20 % and 25 % of $CO_2$ sea-air fluxes measured during July 2017 by the test and standard setup, respectively, were positive. By using the 60 % stationarity limit, only 5 % (test) and 6 % (standard) of the measured $CO_2$ fluxes were positive. With the 30 % stationarity limit,

10  these numbers were only slightly lower, 3 % and 4 %, respectively.

    Similarly high rejection rates have been obtained in previous studies. Miller et al. (2010) reported that 65 % of the sea-air $CO_2$ flux measurements failed a stationarity test in which the 13.7 min fluxes were compared with the averages of 3.4 min fluxes, and assumed that $CO_2$ concentration heterogeneity could be a reason for the high rejection rate. Likewise, Blomquist et al. (2014) concluded that horizontal $CO_2$ concentration gradients caused by continental pollution sources can reduce the





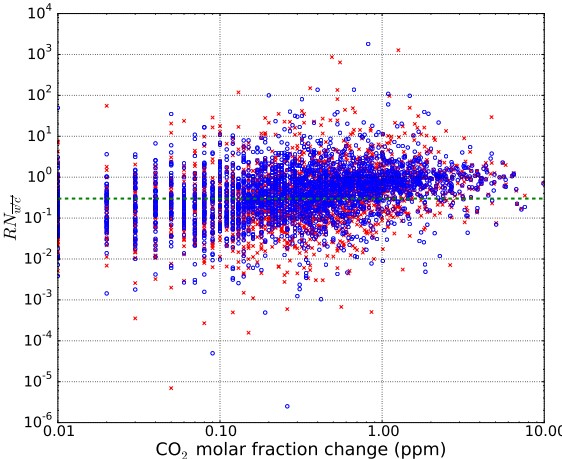

**Figure 7.** Relative non-stationarity of $CO_2$ flux as a function of the absolute change in $CO_2$ concentration during the averaging period: standard (red cross) and test (blue circle) setups. The green dashed line indicates the limit of $RN_{\overline{w'c'}} = 0.3$.

number of stationary situations. At Utö, concentration gradients may be produced by the horizontal heterogeneity due to the land-sea interface and the variations in marine primary production.

We examined how the absolute change in $CO_2$ concentration during the averaging period of 30 min relates to non-stationarity of $CO_2$ fluxes (Fig. 7b). The change in $CO_2$ concentration was calculated as a difference in the mean $CO_2$ molar fraction of the

5 last 30 s and the first 30 s. Typically, a change larger than 1 ppm was associated with rejection of the measurement due to the non-stationarity of $CO_2$ flux. Only 14 % and 18 % of the sea-air $CO_2$ fluxes for the test and standard setup, respectively, passed the stationarity test with the 30 % limit, for the cases with a $CO_2$ concentration change larger than 1 ppm. For the cases where $CO_2$ concentration changed more than 2 ppm during the averaging period, only 5 % and 6 % of sea-air $CO_2$ fluxes passed this stationarity test.

If the time scale of the processes generating non-stationarity was less than the averaging time, it would be possible to reduce the amount of discarded non-stationary data by shortening the averaging time. To test for non-stationarity with shorter averaging periods, we calculated $CO_2$ fluxes for 15 min periods, which were compared with the average of 2.5 min $CO_2$ fluxes. It was found that this increased the number of accepted data only a little: by 6.1 % for the standard setup and 1.9 % for the test setup.

Even though a large number of the measurements did not meet the stationarity criterion ($RN_{\overline{w'c'}} < 0.3$), the occurrence of non-stationary conditions was temporally random, whereas unsuitable wind directions produced long continuous gaps in the measurement time series. For the calculation of a $CO_2$ budget, these gaps must be filled, for instance by using the measured $pCO_2$ difference and a gas transfer velocity parameterized as function of wind speed. Thus, continuous measurements of surface seawater and atmospheric $CO_2$ concentration provide useful additional data for a sea-air $CO_2$ flux station.



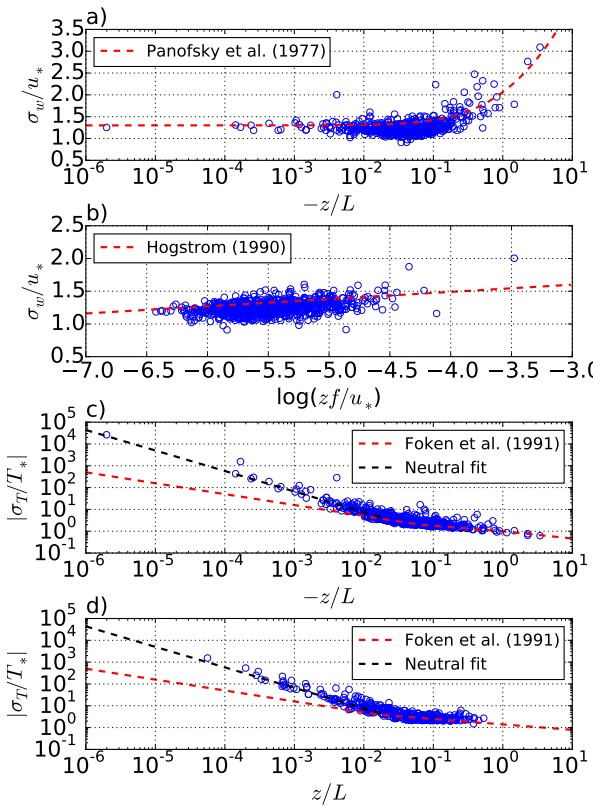

**Figure 8.** Integral turbulence characteristics. (a) $\sigma_w/u_*$ as a function of stability parameter $z/L$, (b) $\sigma_w/u_*$ as a function of $z f_c/u_*$ and (c) $\sigma_T/T_*$ as a function of $z/L$ during unstable stability and (d) $\sigma_T/T_*$ as a function of $z/L$ during stable stability.

### 3.1.5 Turbulence

The Monin-Obukhov similarity theory predicts that statistical turbulence variables within the atmospheric surface layer are unique functions of the stability parameter ($z/L$), which combines information about the height above ground, surface shear stress, surface heat flux and buoyant processes. These so-called integral turbulence characteristics have been found to have the same stability dependence over the sea as on land (e.g. Smith and Anderson, 1984) and can be used to examine the development of turbulence.

Our data are in accordance with the results of Panofsky et al. (1977), who determined the relationship between the normalized standard deviation of vertical velocity and stability parameter:

$$\frac{\sigma_w}{u_*} = \alpha \cdot (1 - \beta \cdot z/L)^{1/3} \tag{6}$$





where $u_*$ is friction velocity, $\alpha = 1.3$ and $\beta = 3.0$. With large $-z/L$ values, the power law of 1/3 fits well to our observations, and the observed $\frac{\sigma_w}{u_*}$ approaches a constant of 1.3 in the neutral range similarly to Eq. 6. Most of the observations fall into the stability range between -0.1 and -0.01. Within this range, the observed $\frac{\sigma_w}{u_*}$ is mainly scattered between values 1.0 and 1.5.

However, Monin-Obukhov similarity theory does not consider all the necessary information for describing turbulence char-
acteristics in all conditions; e.g. the effect of Coriolis and pressure-gradient forces are excluded. Högström (1990) showed that during neutral stratification $\frac{\sigma_w}{u_*}$ can be described with $z$, $u_*$ and Coriolis parameter, $f_c$. Smedman (1990) found that this relationship is valid at widely differing sites, including both terrestrial and marine surfaces. Thus we tested our data with neutral stratification, $|z/L| < 0.1$, against the function proposed by Högström (1990):

$$\frac{\sigma_w}{u_*} = \gamma \cdot \log(z f_c / u_*) + \delta, \tag{7}$$

where $\gamma = 0.11$ and $\delta = 1.93$. This function provides a reasonable fit to our data and a better description of $\frac{\sigma_w}{u_*}$ in the stability range between -0.1 and -0.01 than Eq. 6 (Fig. 8).

For temperature, we compared the observations to the model of Foken et al. (1991):

$$\frac{\sigma_T}{T_*} = \begin{cases} 1.0 \cdot \left(\left|\frac{z}{L}\right|\right)^{-1/3} & , & \frac{z}{L} \leq -1 \\ 1.0 \cdot \left(\left|\frac{z}{L}\right|\right)^{-1/4} & , & -1 \leq \frac{z}{L} \leq -0.0625 \\ 0.5 \cdot \left(\left|\frac{z}{L}\right|\right)^{-1/2} & , & -0.0625 \leq \frac{z}{L} \leq 0.02 \\ 1.4 \cdot \left(\left|\frac{z}{L}\right|\right)^{-1/4} & , & \frac{z}{L} \geq 0.02 \end{cases} \tag{8}$$

Our observations were in accordance with the shape of this relationship, except for the near-neutral stability range, where the
observations differ greatly from the model (Fig. 8c-d). As these cases have a very small sensible heat flux, $\frac{\sigma_T}{T_*}$ could be biased by division by a small number. However, this is not observed as the slope is well-organized and the spread is small. Our data suggest a steeper relationship should be used in this near-neutral stability range ($|z/L| < 0.02$):

$$\frac{\sigma_T}{T_*} = 0.1 \cdot \left(\left|\frac{z}{L}\right|\right)^{-0.94} \tag{9}$$

In the logarithmic scale, the slope (-0.94) in our fit is almost twice as steep as the corresponding slope determined by Foken et
al. (1991), i.e. -0.5.

Overall, our observations were well organized as a function of stability parameter, or friction velocity in the neutral case, and thus we did not discard any of data based on the integral turbulence characteristics.

## 3.2   Setup comparison

The observed direction of the sea-air $CO_2$ fluxes was mainly consistent with the $pCO_2$ difference between the sea and the
atmosphere, and both setups showed similar fluxes most of the time (Fig. 9). The standard setup tended to show slightly more positive fluxes than the test setup. In July, the $pCO_2$ difference was negative, and the mean sea-air $CO_2$ flux measured with the standard and test setup was -0.158 and -0.231 $\mu\mathrm{mol}\,\mathrm{m}^{-2}\,\mathrm{s}^{-1}$, respectively. During the latter part of August (15–31



August), when the $pCO_2$ difference was shifting from negative to positive, the mean flux measured with the standard system ($0.003\,\mu\mathrm{mol\,m^{-2}\,s^{-1}}$) was close to zero, while the test system still showed clearly negative fluxes (mean $-0.082\,\mu\mathrm{mol\,m^{-2}\,s^{-1}}$). During this period, the highest latent heat fluxes of the measurement period were measured, peaking at $274\,\mathrm{W\,m^{-2}}$. During September, low latent heat fluxes ($16\,\mathrm{W\,m^{-2}}$ on average) were observed and both measurements showed the same mean

monthly sea-air $CO_2$ flux: $0.180\,\mu\mathrm{mol\,m^{-2}\,s^{-1}}$. The highest absolute monthly $CO_2$ fluxes were measured in October, when $pCO_2$ difference was continuously high ($16.6\,\mathrm{Pa}$ on average) and wind speed peaked occasionally, resulting in monthly sea-air $CO_2$ flux averages of $0.364$ and $0.301\,\mu\mathrm{mol\,m^{-2}\,s^{-1}}$ by the standard and test setup, respectively.

There is limited amount of direct $CO_2$ flux measurements from this part of the Baltic Sea. The magnitude of the measured monthly sea-air $CO_2$ fluxes was of the same order as the modeled sea-air $CO_2$ fluxes in the Eastern Gotland basin (Norman

et al., 2013). The modeled sink strength during the summer months was $-0.1\,\mu\mathrm{mol\,m^{-2}\,s^{-1}}$, whereas the measured flux was approximately twice as high in Utö in July. The shift from the negative to positive $pCO_2$ difference in Utö occurred two months before the corresponding shift in the model run.

The Nafion drier eliminated the water vapor fluctuations effectively. The latent heat flux (measured with the standard system) was mostly positive, ranging within $-31$–$297\,\mathrm{W\,m^{-2}}$, with an average of $53\,\mathrm{W\,m^{-2}}$. The $H_2O$ variance measured

with the standard setup was $0.042\,\mathrm{mmol^2\,mol^{-2}}$ on average, whereas the test setup measured the mean $H_2O$ variance of $0.001\,\mathrm{mmol^2\,mol^{-2}}$. Even though the Nafion drier in our setup did not remove water vapor completely, it attenuated the fluctuations at all frequencies, in the same way as the tubing of a closed-path system attenuates temperature fluctuations. Water vapor molar fraction measured by the test setup varied within $2.4$–$8.8\,\mathrm{mmol\,mol^{-1}}$, whereas the standard setup showed water vapor values of $3.8$–$15.6\,\mathrm{mmol\,mol^{-1}}$. Thus, a total removal of water vapor from the sample air is not required to eliminate

the water vapor fluctuations.

A high correlation was found between the two measurement setups (Fig. 10). The Pearson product-moment correlation coefficient between the sea-air $CO_2$ fluxes measured by the setups was 0.96. During low latent heat fluxes ($< 30\,\mathrm{W\,m^{-2}}$), the correlation coefficient increased to 0.99. As latent heat flux increases, the standard setup shows more positive $CO_2$ fluxes than the test setup (Fig. 11). These results are in agreement with the conclusions of Landwehr et al. (2014), who found that during

low latent heat fluxes the difference between the dried and undried sea-air $CO_2$ flux measurements were very similar, also proving that a Nafion drier (in our case combined with a virtual impactor) does not disturb the $CO_2$ flux measurements.

A comparison of the $CO_2$ flux difference as a function of latent heat flux shows that the error is negligible for very small latent heat fluxes ($< 10\,\mathrm{W\,m^{-2}}$). With high latent heat fluxes ($\sim100\,\mathrm{W\,m^{-2}}$), the mean difference is approximately $0.15\,\mu\mathrm{mol\,m^{-2}\,s^{-1}}$. Since the sea-air fluxes of $CO_2$ are typically small, the effect of water vapor on $CO_2$ flux measurement

can cause a change in its sign, as observed during the latter part of August. However, as the mean difference increases, also the scatter increases. There are only few data points with negative latent heat fluxes, so no conclusions can be drawn concerning the effect of negative water vapor fluxes on $CO_2$ fluxes.

We showed that the use of the combination of a virtual impactor and a Nafion drier did not disturb the measurement of $CO_2$ fluxes. Closed-path instruments are typically protected by one or two Teflon filters (one close to the inlet and another next to

the instrument), which should prevent any liquid water from reaching the instrument. However, the use of a Teflon filter close





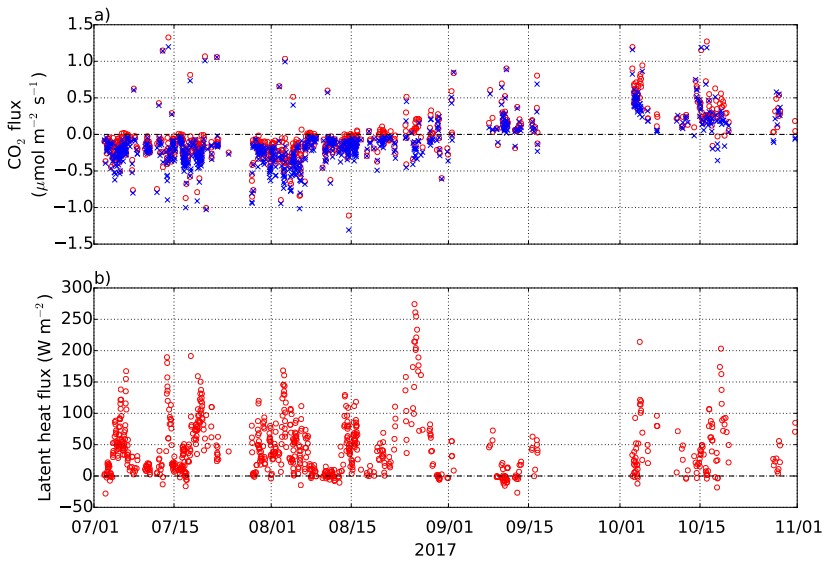

**Figure 9.** Sea-air fluxes of (a) $CO_2$ and (b) water vapor: standard setup (red circles) and test setup (blue crosses). Only the fluxes originating from the sea sector (Fig. 1) are shown.

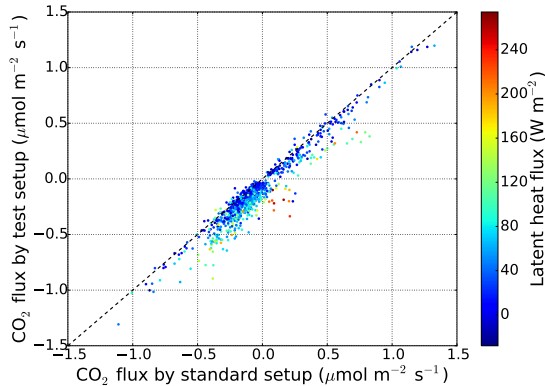

**Figure 10.** The sea-air $CO_2$ flux measured with the standard setup vs. the test setup (equipped with a drier and a virtual impactor). The marker color indicates the concurrent latent heat flux, and the dashed line represents the 1:1 relationship.

to the inlet may be unpractical, as a regular change of this filter may prove laborious. In such a case, we suggest a virtual impactor as an option for the protection of the instrument and the tubing from water and sea salt.




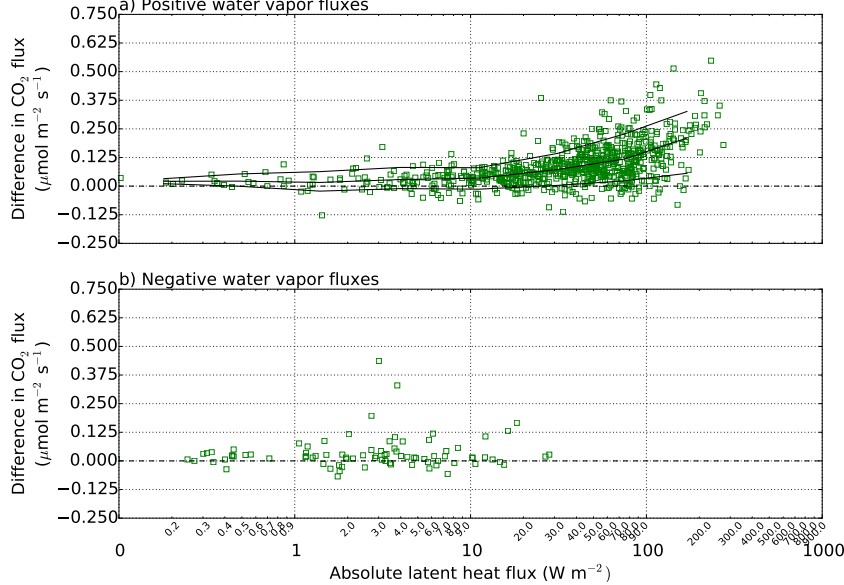

**Figure 11.** $CO_2$ flux difference as a function of latent heat flux (standard setup – test setup) for (a) positive and (b) negative water vapor fluxes. The lines show a fitted mean together with the 10th and 90th percentiles for eight logarithmically equally spaced bins.

## 4  Conclusions

In this paper, we presented new sea-air $CO_2$ flux measurements, comparing two closed-path gas analyzer setups installed on a shore of an island in the Baltic Sea. One of the setups was equipped with a drier and a virtual impactor. By inspecting several theoretical assumptions of the eddy covariance method, we showed that this land-based flux site is capable of effec-
tively monitoring $CO_2$ fluxes between the atmosphere and the sea. The $w$-$CO_2$ cospectral densities calculated from our data showed an expected behavior with a moderate high frequency loss. Turbulence was well-developed, and the integral turbulence characteristics followed unique functions of stability and friction velocity.

We found that the two closed-path infrared gas analyzer setups showed similar $CO_2$ fluxes when latent heat fluxes were small. Correlation coefficient increased from 0.96 to 0.99 when only the $CO_2$ fluxes during small latent heat fluxes ($< 30\,\mathrm{W\,m^{-2}}$)
were included. A higher latent heat flux resulted in a positive bias to the undried measurement. During high latent heat fluxes (~$100\,\mathrm{W\,m^{-2}}$), the difference between the $CO_2$ flux measurements was $0.15\,\mathrm{\mu mol\,m^{-2}\,s^{-1}}$ on average, which is comparable to the monthly mean sea-air $CO_2$ flux. In July, the $CO_2$ partial pressure in the atmosphere exceeded that in the surface seawater, resulting in negative sea-air $CO_2$ fluxes, with monthly averages of -0.16 and -0.23 $\mathrm{\mu mol\,m^{-2}\,s^{-1}}$ for the standard and test (drier) setup, respectively. In October, when the surface seawater had a higher $CO_2$ partial pressure than the atmosphere, the
average sea-air $CO_2$ fluxes were 0.36 and 0.30 $\mathrm{\mu mol\,m^{-2}\,s^{-1}}$ for the standard setup and test setup, respectively.



Even though producing a high percentage of discarded observations (55–63 %), for high quality sea-air $CO_2$ flux measurements it is essential to reject non-stationary observations. If the non-stationary cases were not discarded, 20–25 % of the sea-air $CO_2$ fluxes in July had a wrong sign, whereas only 3–4 % of the measured fluxes had an unphysical direction if the non-stationary cases were rejected. This non-stationarity was found to be linked to the changes in atmospheric $CO_2$ concen-

tration during the averaging period. A change larger than 2 ppm was associated with in the rejection rate of 94–95 % due to non-stationarity.

We showed that the use of the combination of a virtual impactor and a Nafion drier did not disturb the $CO_2$ flux measurement. While this configuration generated a slightly higher flux loss, we opt for this alternative for its reduced water vapour cross-sensitivity and better protection against sea spray.

**Appendix A: Virtual impactor**

The virtual impactor is based on two perpendicular air streams, which provides a means to separate particles by size (Fig. A1). The sample air stream is divided into minor and major flows : smaller particles are diverted to the major flow and large particles with higher inertia, in this case water droplets, continue to the minor flow. The 50 % cut-point of our virtual impactor was calculated to be 1.2–1.3 µm, i.e. a particle of this size has a 50 % probability of removal, while smaller particles are more

likely to enter the major flow.

*Competing interests.* The authors declare that they have no conflict of interest.

*Acknowledgements.* This work was financially supported by the Finnish Meteorological Institute, the BONUS INTEGRAL project (BONUS Blue Baltic) and the JERICO-NEXT project (EU - Horizon 2020, 654410). The Integrated Carbon Observation System is acknowledged for providing the data of atmospheric $CO_2$ concentrations in Utö.



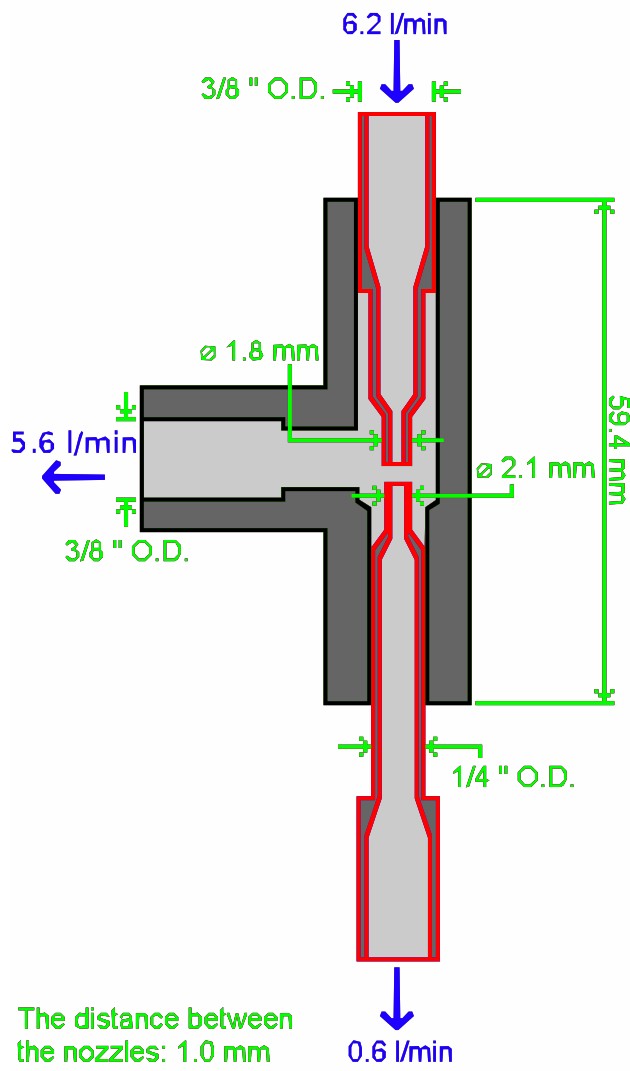

**Figure A1.** Schematics of the virtual impactor attached to the test setup. O.D. stands for outside diameter. The red pipes are enclosed by the black outer case. A more detailed figure is available on request.

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
