# Peer review of "Measuring turbulent $\mathrm{CO}_2$ fluxes with a closed-path gas analyzer in marine environment"

_Atmospheric Measurement Techniques, 2018_

## Referee Comment (RC1) · G. Bohrer (Referee) · 1 May 2018

The manuscript provides a very technical assessment of the effects of a drier and a compactor on removing the spectral cross contamination of water vapor in co2 flux measurements. It is relevant, and reports interesting findings with direct implications to observation techniques of ocean co2 fluxes. Given the topic, I found the narrative diverging to describing other onrelated findings (turbulence, gap filling, roughness lenght). These sections should be removed, or, the direct relevance of these findings to understanding the differences between the measurement setups should be described.

otherwise, I only have a few minor comments: p3L15-20, I think sea breaze is worth mentioning here. Also, you should mention specifically

[Figure]

that sea-land gradient is almost always present and violates the horizontal homogeneity assumption. please reference Rey Sanchez et al 2017 https://www.tandfonline.com/doi/abs/10.1080/20964129.2017.1392830 which evaluated the relative effect of horizontal affection.

Fig 1- please say explicitly where is the ec flux station. I assume it is A, but a bit confused about what B is.

the location of the tower at the edge of a cliff is problematic. The sharp change of roughness and the physical disturbance to the flow probably generate increased turbulence and a vertical ejection flow that violate a few of the ec assumpsions (0 mean vertical flow, ergodicity of turbulence). There are many papers discussing the effects of forward facing step on vertical flow. See for example our paper https://link.springer.com/article/10.1007%2Fs10546-014-9923-5 and references within. Roughness and surface heat flux transition create circulations patterns that are particularly problematic for edge-of-shore flux measurements. See Higgins et al 2013 https://journals.ametsoc.org/doi/full/10.1175/JHM-D-12-0166.1 and Kenny at al 2017 https://link.springer.com/article/10.1007/s10546-017-0268-8 . these will be a problem even when the footprint is all water. Please at least discuss this issue, around where you discuss other difficulties of measuring carbon flux in the ocean. P7L15 in the same issue of vertical flow - how did you rotate your wind coordinates? Most rotations assume 0 vertical wind. That will affect your momentum flux. In your case with the cliff facing the wind, you should filter out cases where the unrotated mean w was large. P7L26 you cannot remove more than half of the observations based on a subjective eyeball analysis for "distortion". Please provide an objective definition for which observations should be removed and remove only and all of those that fit these criteria. P12 L5 The stationarity is an environmental property and should not vary between instruments. I do not understand why you have different stationary cases for each of your setups which are at the same location. That can only be the outcome of different observation errors in each of the setups. I suggest adopting a common criterion, when both

sensors observe stationary conditions. P13L19 "co2 flux station" seems odd. maybe "co2 flux exchange at the station". This paper is not about carbon budgets. I suggest removing this section. if you insist to keep it, please provide moreinformation on the gap filling approach you used, how you estimated uncertainty, and the resulting co2 flux budget.

---

## Referee Comment (RC2) · B. Butterworth (Referee) · 19 Jun 2018

General Comments

The work laid out in this methods paper is thorough and up-to-date, with the current best practices in the field. The researchers designed and clearly described what appears to be a functional eddy covariance tower for measuring CO2 fluxes in a marine environment. Which is not an easy task. With regard to scientific quality, I believe the authors have done an excellent job.

There are several areas that the paper could be improved. Broadly, the paper needs to stay more focused on CO2 flux measurements. Several analyses seemed unnecessary (e.g., roughness length and similarity theory), veering away from the overriding

goal. These could be replaced by a more in-depth assessment of the performance of the system with regard to CO2 flux (e.g., gas transfer velocity or bulk flux comparisons). Also, because the method (i.e., dried closed-path IRGA) has been presented in previous papers, this paper would improve its contribution by delving a little deeper into the challenges facing the system (e.g., distorted spectra). Doing so could enable those deploying similar systems in the future to address those challenges and improve data retention.

This is a good paper. The work is thorough, the methods are sound. I recommend it for publication and look forward to seeing the science papers that follow.

Specific comments

The paper should attempt some way to corroborate the magnitude of the measured fluxes with previous results. While we have no reason to doubt that the fluxes are good, we also have no evidence that they are good. Accomplishing this should be straightforward given the fact that waterside pCO2 was measured. This could be done by presenting gas transfer velocity. Or it could be done by comparing against bulk CO2 flux calculated using an existing parameterization for gas transfer velocity.

The percentages of data lost due to non-stationarity (63%) and wind direction (51%) are reported separately. It's worth reporting somewhere in the paper the combined loss / total percentage of data that made it to the final analysis.

Page 3 – line 13 –The closed-path design does not automatically mean increased sensitivity to motion. Miller et al. (2010) found that the LI6262 and LI7000 were more sensitive to motion. This doesn't apply to all closed-path IRGAs on the market (e.g., the LI7200 [closed-path], which has the same internal design as a LI7500 [open-path], doesn't show the same degree of motion sensitivity as those named by Miller et al. [2010])

Page 5 – I like the photograph (Figure 2). It makes the case that the virtual impactor

is necessary. But it seems like a photograph that is a little closer to the tower, and that shows the instrumentation a little better, would be more helpful here (where you're describing the physical design of the system) and that this one could potentially be moved to the Appendix. Just a thought.

Page 6 – How do you generate your purge air for the nafion? And zero gas for the LI7000?

Page 7 – line 14 – Suggest removing the "flow" in "stationary flow conditions". It's not purely flow that is involved (w'c').

Page 7 – Section 2.3.1. Spectral analysis:

Here you've applied a limit of |z/L| < 0.05. But later, in Figure 8, it appears that about half of the data fall outside of this range. Why did you apply this more stringent criteria? Was there something wrong with the spectra outside this range? Or was it just so the peaks of your spectra lined up more cleanly?

Within the more limited selection of 612 observations, over half are discarded because they are 'distorted.' I assume this is so you can use the 'good' spectra to get a reasonable/workable transfer function. That makes sense. But then are you applying that transfer function to correct all observations that pass the stationarity and wind direction criteria, even the ones with distorted spectra? The transfer function should not be expected to fix the fluxes for these other intervals.

I think this is an area of the paper that can be developed. If the quality control for CO2 fluxes (shown in Figures 6 and 9) consists only of the wind direction and stationarity criteria, then the distortion of spectra mentioned in this section needs to be addressed. What is causing the 365 of 612 intervals that satisfy |z/L| < 0.05 (and presumably some percentage of intervals with z/L outside this range) to have distorted spectra?

Is the distortion in the spectra from low or high frequencies? Or a consistent spike? Does it happen under specific environmental conditions? Is there some way to measure and account for the error such distortions likely introduce into the measured flux? The shape of the spectra, especially if they're consistent, will be useful for indicating what problems may be affecting the tower. Maybe it's low frequency contribution coming from the residual water vapor, maybe the tower is swaying, maybe there is high frequency noise from the IRGAs. If you can give the readers more information here it will help those who plan to deploy similar systems to address those challenges.

Page 9 – Figure 3 – When printed the dashed cyan line on subplots d and e is not visible when it overlaps with the blue line. Might want to consider some way to increase the contrast.

Page 10 – Section 3.1.2 – With respect to z0, no one was going to complain that an extended open water surface does not satisfy horizontal homogeneity conditions for EC measurements. Yes, the figure does show that you've successfully selected the right wind direction window. But that seemed obvious just from your map. And you could always just include one sentence that says it was confirmed because z0 was consistently below 1 mm in that window. There isn't anything wrong with this section, but it is not particularly necessary.

Page 7 – line 3 – Here you've handled the tube delays by taking the maximum co-variance. I know this is a common practice. But my experience has been that it often results in the selection of fluxes which have large contributions from frequencies outside the expected range. Have you calculated the expected tube delays based on your known tube lengths, tube diameters, and flow rate? How well do the maximum co-variance lags match the expected lag based on this calculation? If you base the tube delay on this calculation do you see improvement in the percentage of spectra that are distorted (page 7 – line 26)? I don't think this necessarily needs to be reported on in the final paper, but it's worth checking.

Page 11 – line 2 – What is the average high-frequency correction?

Page 11 – line 15 – How do you handle the range of f0 (e.g., 0.04 to 0.22) in correcting

the attenuation of latent heat flux? Did you calculate the different transfer functions for different relative humidities based on their $f_0$?

Your assertion that the flux attenuation (for RH=80%) is 44% of the real flux would be strengthened by a comparison of your result to an empirically-based model (e.g., COARE [Fairall et al. 2003]).

Not critical, but since you are using these latent heat fluxes to interpret your $CO_2$ fluxes it wouldn't hurt to show the CowH2O cospectra of the undried system.

Page 12 – line 8 – So the reader doesn't have to flip back several pages to Figure 3 it would be helpful here to place a reminder that all the $CO_2$ fluxes in July were expected to be negative. Maybe insert something like " (all of which were expected to be negative) " after "July 2017".

Page 14 – Section 3.1.5 Turbulence: I think this section is unnecessary. It distracts from the main purpose of the paper ($CO_2$ fluxes). In reading it felt like a distinct transition to a new subject. One which required more effort than the payoff was worth. I think this section would be better left to its own paper, when the authors have developed it further and have the space to discuss the implications. For this paper, there was no practical application. If there is a reason that this relates to the $CO_2$ fluxes that you've shown then that reason should be made more clear.

Technical Corrections

Page 3 – line 26 – Word choice on "effortless". Might want to go with something like more feasible, practical, or straightforward (or conversely 'not as logistically challenging').

Page 3 – line 31 – Grammar: "and the drying of sample is straightforward to implement" to "and allows for straightforward implementation of sample air drying" (or similar)

Page 5 – Citations at the ends of each paragraph should be moved inside the period.

Page 5 – line 17 – Change "A . . . tower is placed" to "A . . . tower was placed"

Page 7 – line 6 – "was considered" to "were considered"

Page 7 – line 23 – "based on the criterion that RNw'c'<0.3" would read better if it were enclosed in parentheses rather than commas. This would match the way you've presented the wind direction criterion.

Page 8 – line 3 – Consider reorganizing the sentence that begins with "To correct for...". It is difficult to read.

Page 8 – line 26 – "seasons of carbon cycle" to "seasons of the carbon cycle"

Page 10 – line 13 – "Especially, a large swell . . ." is an incomplete sentence.

Page 14 – Figure 8 – "f" in the xlabel of subplot b should be "fc" for consistency with how it's written in the text.

Page 16 – line 25 – "the difference between the dried and undried sea-air CO2 flux measurements were very similar" needs to either become "the difference between the dried and undried sea-air CO2 flux measurements was small" or "the difference between the dried and undried sea-air CO2 flux measurements were very similar."

Also, the second part of this sentence doesn't work as it's written because the subject of the sentence changes.

Page 18 – Figure 11 caption – move "(standard setup – test setup)" to immediately after "CO2 flux difference"

Throughout the text hyphens are used for minus signs.

References

Blomquist, B. W., Huebert, B. J., Fairall, C. W., Bariteau, L., Edson, J. B., Hare, J. E. and McGillis, W. R.: Advances in Air-Sea CO2 Flux Measurement by Eddy Correlation, Bound.-Lay. Meteorol., 152, 245–276, doi:10.1007/s10546-014-9926-2, 2014.

Fairall, C. W., Bradley, E. F., Hare, J. E., Grachev, A. A., and Edson, J. B.: Bulk parameterization of air-sea fluxes: updates and verification for the COARE algorithm, J. Clim., 16, 571–591, 2003.

Miller, S. D., Marandino, C. A. and Saltzman, E. S.: Ship-based measurement of air-sea CO2 exchange by eddy covariance, J. Geophys. Res., 115, D02304, doi:10.1029/2009JD012193, 2010.

---

## Author Comment (AC1) · 24 Jul 2018

We thank the Referee for his comments. Answers, together with the revised manuscript, are given in the supplement.

Please also note the supplement to this comment:
https://www.atmos-meas-tech-discuss.net/amt-2018-61/amt-2018-61-AC1-supplement.zip

---

## Editor Decision (ED1)

**Editor Comments**
to revised manuscript of Honkanen et al. (amt-2018-61)

**NOTE:** The following page (P) and line (L) numbers refer to the revised manuscript version with track changes.

1) Response to RC1.6:
The author response is satisfying but I think it would be important to include the observations about the vertical rotation angles in the manuscript text.
"The vertical rotation angle within the sea sector varied within 0-5° with a mean of 3.1° and a standard deviation of 1.4°, indicating that the flow divergence due to the cliff is limited."

2) P3 L2: Why should it not be possible with open-path analysers to correct the dilution due to water vapor as a point-by-point operation on the high frequency data? The open-path sensors make synchronous fast-response measurements of $CO_2$ and $H_2O$!
Instead, it could be pointed out that the density variations due to temperature fluctuations cannot be corrected on the high frequency data of an open-path sensor due to its separation from the sonic.

3) P8 L21: "the outermost points were discarded". Please explain, why and on what scientific basis this has been done.

4) P9 L5: If the seawater inlet was attached to a float, the statement "…is kept 4.5 m below the mean sea level" is erroneous. The mean sea level is a geographical/cartographic altitude term and does not account for e.g. tidal changes. In my understanding the sentence should be rephrased to "…is kept 4.5 n below the water surface"

5) Section 2.4: The focus of the present manuscript is on $H_2O$ cross-sensitivity effects of the $CO_2$ exchange measurements. And it is argued in the Conclusions, that the monitoring of the water-air $CO_2$ gradient provides useful additional information (e.g. for gap filling).
In the light of this argumentation, the authors should also address the problem of $H_2O$ cross-sensitivity in the water-air gradient measurements as described in Section 2.4. Apparently the pCO2-eq concentration was not measured with pre-drying of the samples? Here the problem of cross-sensitivity might be even larger, because the water and air concentration are carried out with two different instruments.

LANGUAGE CORRECTIONS (list is not exhaustive!)

P5 Figure 1 Caption: replace "flow-through pumping system" with "sea water sampling system" to be more specific (see also Comment to P8 L32 below)

P7 L23: Correct to: "Here, a 30 min period …"

P8 L12: Correct to: "…met this criteria."

P8 L15: Correct to: "…within a predefined range …"

P8 L22: Taking into account the revisions in the preceding sentences, this line should be rephrased to: "…the transfer function in Eq. (4) was fitted to …"

P8 L32: The expression "flow-through pumping system" is not very specific and needs clarification (In my understanding, each pump is a flow-through system!). I guess an "immersion pump" or "submersible pump" was meant here?

P9 L4: Omit "intake" here (redundant with "inlet" in the same sentence)

P10 Figure 3 caption (4th line): change "cyan" to "orange", like in the previous line.

---

## Author Response (AR2)

**Response to editor comments**

Authors reply to each comment in the following. Black text indicates editor comments and the green text is authors' response. Page and line numbers refer to the most recently revised manuscript. In addition to the following changes, we also checked the whole manuscript for correct phrasing.

We also found out that we were luckily able to recover additional temperature measurement of the inlet (from a standalone CTD next to the inlet) for the beginning of the measurement period, so the Fig. 3 in the revised manuscript also includes sea water temperature at 4.5 m depth and temperature corrected pCO2 for the beginning of the period.

Editor Comments

*to revised manuscript of Honkanen et al. (amt-2018-61)*

*NOTE: The following page (P) and line (L) numbers refer to the revised manuscript version with track changes.*

*1) Response to RC1.6:*

*The author response is satisfying but I think it would be important to include the observations about the vertical rotation angles in the manuscript text. "The vertical rotation angle within the sea sector varied within 0-5° with a mean of 3.1° and a standard deviation of 1.4°, indicating that the flow divergence due to the cliff is limited."*

Authors agree that this is an important result for the general quality assurance. A sentence was added to the P9L24:

> The vertical rotation angle within the sea sector during the measurement period varied within 0–5 ° with a mean 3.1 ° and a standard deviation of 1.4 °, indicating that the flow divergence due to the cliff is limited.

*2) P3 L2: Why should it not be possible with open-path analysers to correct the dilution due to water vapor as a point-by-point operation on the high frequency data? The open-path sensors make synchronous fast-response measurements of CO2 and H2O! Instead, it could be pointed out that the density variations due to temperature fluctuations cannot be corrected on the high frequency data of an open-path sensor due to its separation from the sonic.*

Authors have rephrased this in the revised manuscript. P2L32 states now:

> For both analyzer types, the dilution due to water vapor (Webb et al., 1980) can be corrected accurately as a point-by-point operation on the high frequency data (Ibrom et al., 2007). However, such an approach is not feasible for open-path sensors, because the density variations due to temperature cannot be compensated for due to the spatial separation between the $CO_2$ and temperature sensor.

*3) P8 L21: "the outermost points were discarded". Please explain, why and on what scientific basis this has been done.*

The reason for discarding points is now clarified in the revised manuscript (P8L7):

> *Also, 4 outermost points from the high frequency end were discarded because these points may easily be biased due to the division by very small number and 20 outermost points from the low frequency end were discarded because these points do not play role in the high frequency attenuation.*

*4) P9 L5: If the seawater inlet was attached to a float, the statement "…is kept 4.5 m below the mean sea level" is erroneous. The mean sea level is a geographical/cartographic altitude term and does not account for e.g. tidal changes. In my understanding the sentence should be rephrased to "…is kept 4.5 n below the water surface"*

The sentence is now rephrased to include sea level changes (P8L23):

> The bottom-moored, floating inlet for seawater is located 250m west of the station (Fig. 1) and is kept approximately 4.5m below the mean sea level: the sampling depth typically varies within 4.0–5.0m.

During the measurement period, the water level was close to the theoretical mean sea level, and thus, during this period the depth of inlet was approximately 4.5 m.

*5) Section 2.4: The focus of the present manuscript is on H2O cross-sensitivity effects of the CO2 exchange measurements. And it is argued in the Conclusions, that the monitoring of the water-air CO2 gradient provides useful additional information (e.g. for gap filling). In the light of this argumentation, the authors should also address the problem of H2O cross-sensitivity in the water-air gradient measurements as described in Section 2.4. Apparently the pCO2-eq concentration was not measured with pre-drying of the samples? Here the problem of cross-sensitivity might be even larger, because the water and air concentration are carried out with two different instruments.*

Authors do not see the cross-sensitivity as a marked problem for the absolute concentration measurements (P8L19):

> Since this $CO_2$ measurement is not carried out using pre-dried sample air, a $H_2O$ cross-sensitivity effect on $CO_2$ is possible but likely small compared to the other error sources, such as spatial heterogeneity in sea water $CO_2$ concentration.

*LANGUAGE CORRECTIONS (list is not exhaustive!)*

*P5 Figure 1 Caption: replace "flow-through pumping system" with "sea water sampling*

*system" to be more specific (see also Comment to P8 L32 below)*

Fig 1 caption on page 4 is now rephrased:

Figure 1. Location of the Utö island in the Archipelago Sea. The research installations on the island consist of the Utö Atmospheric and Marine Research Station and its flux tower (A), the inlet of the sea water sampling system…

*P7 L23: Correct to: "Here, a 30 min period …"*

Done. P7L14:

Here, a 30 min period is regarded as stationary if both of the CO2 flux measurements fulfil the Condition…

*P8 L12: Correct to: "…met this criteria."*

Done. P7L29:

607 half-hour periods met these criteria.

*P8 L15: Correct to: "…within a predefined range …"*

Done. P7L30:

Furthermore, a cospectrum was discarded if its peak was not within a predefined range (0.01–0.5 Hz).

*P8 L22: Taking into account the revisions in the preceding sentences, this line should be rephrased to: "…the transfer function in Eq. (4) was fitted to …"*

Done. *P8L5:*

*Using a non-linear least squares fit, the transfer function in Eq. 4 was fitted to the ratio of …*

*P8 L32: The expression "flow-through pumping system" is not very specific and needs clarification (In my understanding, each pump is a flow-through system!). I guess an "immersion pump" or "submersible pump" was meant here?*

Done. P8L16:

In addition to the flux measurements, we present here data of the dissolved CO2 concentration in surface seawater, which is measured using a SuperCO2 system (Sunburst Sensors LLC) connected to a submersible pump.

*P9 L4: Omit "intake" here (redundant with "inlet" in the same sentence)*

Done. P8L23:

The bottom-moored, floating inlet for seawater is located 250m west of the station (Fig. 1) and is kept approximately 4.5m below the mean sea level: the sampling depth typically varies within 4.0–5.0m.

*P10 Figure 3 caption (4th line): change "cyan" to "orange", like in the previous line.*

Done. Fig 3 caption on page 10:

[revised manuscript text omitted]
 closely similar monthly sea-air $CO_2$ flux: 0.1800.188 $\mathrm{\mu mol\,m^{-2}\,s^{-1}}$ (test) and 0.158 $\mathrm{\mu mol\,m^{-2}\,s^{-1}}$ (standard). The highest absolute monthly $CO_2$ fluxes were measured in October, when $pCO_2$ difference was continuously high (16.6 Pa on average) and wind speed peaked occasionally, resulting in monthly sea-air $CO_2$ flux averages of 0.364 and 0.3010.524 and 0.434 $\mathrm{\mu mol\,m^{-2}\,s^{-1}}$ by the standard and test setup, respectively.

There is limited amount of direct $CO_2$ flux measurements from this part of the Baltic Sea. The As no previous measurement data are available for comparison, we validated the magnitude of the measured monthly sea-air air-sea $CO_2$ fluxes was of the same order as the modeled sea-air roughly by calculating the gas transfer velocity ($k_{660} = \frac{F}{K_0 \Delta pCO_2} \sqrt{\frac{Sc}{660}}$) from our data and compared it with the universal parametrization proposed by Wanninkhof (1992). Schmidt number ($Sc$) and solubility ($K_0$) of

[Figure]

**Figure 7.** Measured gas transfer velocities as a function of wind speed. The orange line is the parametrization by Wanninkhof (1992).

$CO_2$ were calculated according to Wanninkhof (1992) and Weiss (1974), respectively. For this comparison, we only included cases in which the absolute partial pressure difference ($\Delta pCO_2$) between the sea and atmosphere was larger than 3  Pa. The gas transfer velocities derived from our measurements

5   were mostly in good accordance with predictions (Fig. 7), which lends credence to our flux measurements. During medium wind speeds (3–10 m s$^{-1}$, the measured $k_{660}$ values line up close to the parametrization. More measurements during low ($< 3$ m s$^{-1}$) and high ($> 10$ m s$^{-1}$) wind speeds are required to evaluate the relationship between windspeed and $k_{660}$ during these windspeeds.

The Nafion drier eliminated the water vapor fluctuations effectively. The latent heat flux (measured with the standard system)

10   was mostly positive, ranging within 28–122 W m$^{-2}$, with an average of 27 W m$^{-2}$. The H$_2$O variance measured with the standard setup was 0.021 
[revised manuscript text omitted]